# Modeling Clutter Perception using Parametric Proto-object Partitioning

**Chen-Ping Yu**
Department of Computer Science
Stony Brook University
cheyu@cs.stonybrook.edu

**Wen-Yu Hua**
Department of Statistics
Pennsylvania State University
wxh182@psu.edu

**Dimitris Samaras**
Department of Computer Science
Stony Brook University
samaras@cs.stonybrook.edu

**Gregory J. Zelinsky**
Department of Psychology
Stony Brook University
Gregory.Zelinsky@stonybrook.edu

## Abstract

Visual clutter, the perception of an image as being crowded and disordered, affects aspects of our lives ranging from object detection to aesthetics, yet relatively little effort has been made to model this important and ubiquitous percept. Our approach models clutter as the number of proto-objects segmented from an image, with proto-objects defined as groupings of superpixels that are similar in intensity, color, and gradient orientation features. We introduce a novel parametric method of clustering superpixels by modeling mixture of Weibulls on Earth Mover's Distance statistics, then taking the normalized number of proto-objects following partitioning as our estimate of clutter perception. We validated this model using a new 90-image dataset of real world scenes rank ordered by human raters for clutter, and showed that our method not only predicted clutter extremely well (Spearman's $\rho = 0.8038$, $p < 0.001$), but also outperformed all existing clutter perception models and even a behavioral object segmentation ground truth. We conclude that the number of proto-objects in an image affects clutter perception more than the number of objects or features.

## 1   Introduction

Visual clutter, defined colloquially as a "confused collection" or a "crowded disorderly state", is a dimension of image understanding that has implications for applications ranging from visualization and interface design to marketing and image aesthetics. In this study we apply methods from computer vision to quantify and predict human visual clutter perception.

The effects of visual clutter have been studied most extensively in the context of an object detection task, where models attempt to describe how increasing clutter negatively impacts the time taken to find a target object in an image [19][25][29][18][6]. Visual clutter has even been suggested as a surrogate measure for set size effect, the finding that search performance often degrades with the number of objects in a scene [32]. Because human estimates of the number of objects in a scene are subjective and noisy - one person might consider a group of trees to be an object (a forest or a grove) while another person might label each tree in the same scene as an "object", or even each trunk or branch of every tree - it may be possible to capture this seminal search relationship in an objectively defined measure of visual clutter [21][25]. One of the earliest attempts to model visual clutter used edge density, i.e. the ratio of the number of edge pixels in an image to image size [19]. The subsequent feature congestion model ignited interest in clutter perception by estimating

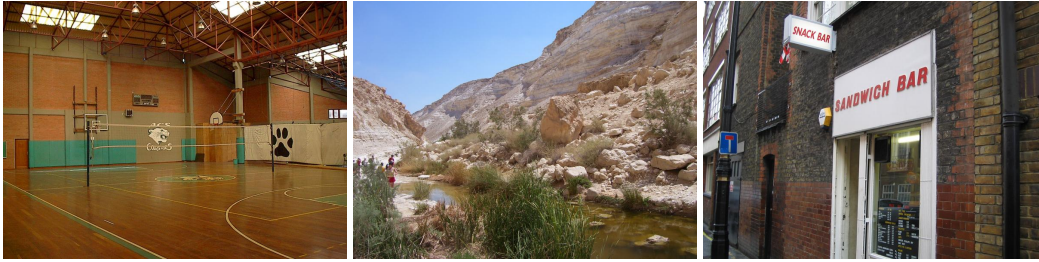

Figure 1: How can we quantify set size or the number of objects in these scenes, and would this object count capture the perception of scene clutter?

image complexity in terms of the density of intensity, color, and texture features in an image [25]. However, recent work has pointed out limitations of the feature congestion model [13][21], leading to the development of alternative approaches to quantifying visual clutter [25][5][29][18].

Our approach is to model visual clutter in terms of proto-objects: regions of locally similar features that are believed to exist at an early stage of human visual processing [24]. Importantly, proto-objects are not objects, but rather the fragments from which objects are built. In this sense, our approach finds a middle ground between features and objects. Previous work used blob detectors to segment proto-objects from saliency maps for the purpose of quantifying shifts of visual attention [31], but this method is limited in that it results in elliptical proto-objects that do not capture the complexity or variability of shapes in natural scenes. Alternatively, it may be possible to apply standard image segmentation methods to the task of proto-object discovery. While we believe this approach has merit (see Section 4.3), it is also limited in that the goal of these methods is to approximate a human segmented ground truth, where each segment generally corresponds to a complete and recognizable object. For example, in the Berkeley Segmentation Dataset [20] people were asked to segment each image into 2 to 20 equally important and distinguishable things, which results in many segments being actual objects. However, one rarely knows the number of objects in a scene, and ambiguity in what constitutes an object has even led some researchers to suggest that obtaining an object ground truth for natural scenes is an ill-posed problem [21].

Our clutter perception model uses a parametric method of proto-object partitioning that clusters superpixels, and requires no object ground truth. In summary, we create a graph having superpixels as nodes, then compute feature similarity distances between adjacent nodes. We use Earth Mover's Distance (EMD) [26] to perform pair-wise comparisons of feature histograms over all adjacent nodes, and model the EMD statistics with mixture of Weibulls to solve an edge-labeling problem, which identifies and removes between-cluster edges to form isolated superpixel groups that are subsequently merged. We refer to these merged image fragments as proto-objects. Our approach is based on the novel finding that EMD statistics can be modeled by a Weibull distribution (Section 2.2), and this allows us to model such similarity distance statistics with a mixture of Weibull distribution, resulting in extremely efficient and robust superpixel clustering in the context of our model. Our method runs in linear time with respect to the number of adjacent superpixel pairs, and has an end-to-end run time of 15-20 seconds for a typical 0.5 megapixel image, a size that many supervised segmentation methods cannot yet accommodate using desktop hardware [2][8][14][23][34].

## 2 Proto-object partitioning

### 2.1 Superpixel pre-processing and feature similarity

To merge similar fragments into a coherent proto-object region, the term fragment and the measure of coherence (similarity) must be defined. We define an image fragment as a group of pixels that share similar low-level image features: intensity, color, and orientation. This conforms with processing in the human visual system, and also makes a fragment analogous to an image superpixel, which is a perceptually meaningful atomic region that contains pixels similar in color and texture [30]. However, superpixel segmentation methods in general produce a fixed number of superpixels from an image, and groups of nearby superpixels may belong to the same proto-object due to the intended over-segmentation. Therefore, we extract superpixels as image fragments for pre-processing,

and subsequently merge similar superpixels into proto-objects. We define that a pair of adjacent superpixels belong to a coherent proto-object if they are similar in *all* three low-level image features. Thus we need to determine a similarity threshold for each of the three features, that separates the similarity distance values into "similar", and "dissimilar" populations, detailed in Section 2.2.

In this work, the similarity statistics are based on comparing histograms of intensity, color, and orientation features from an image fragment. The intensity feature is a 1D 256 bin histogram, the color feature is a $76 \times 76$ (8 bit color) 2D histogram using hue and saturation from the HSV colorspace, and the orientation feature is a symmetrical 1D 360 bin histogram using gradient orientations, similar to the HOG feature [10]. All three feature histograms are normalized to have the same total mass, such that bin counts sum to one.

We use Earth Mover's Distance (EMD) to compute the similarity distance between feature histograms [26], which is known to be robust to partially matching histograms. For any pair of adjacent superpixels $v_a$ and $v_b$, their normalized feature similarity distances for each of the intensity, color, and orientation features are computed as: $x_{n;f} = \text{EMD}(v_{a;f}, v_{b;f})/\widehat{\text{EMD}}_f$, where $x_{n;f}$ denotes the similarity (0 is exactly the same, and 1 means completely opposite) between the $n^{th}$ pair $(n = 1, ..., N)$ of nodes $v_a$ and $v_b$ under feature $f \in \{i, c, o\}$ as intensity, color, and orientation. $\widehat{\text{EMD}}_f$ is the maximum possible EMD for each of the three image features; it is well defined in this situation such that the largest difference between intensities is black to white, hues that are $180°$ apart, and a horizontal gradient against a vertical gradient. Therefore, $\widehat{\text{EMD}}_f$ normalizes $x_{n;f} \in [0, 1]$. In the subsequent sections, we explain our proposed method for finding the adaptive similarity threshold from $\mathbf{x}_f$, which is the EMDs of all pairs of adjacent nodes .

## 2.2 EMD statistics and Weibull distributon

Any pair of adjacent superpixels are either similar enough to belong to the same proto-object, or they belong to different proto-objects, as separated by the adaptive similarity threshold $\gamma_f$ that is different for every image. We formulate this as an edge labeling problem: given a graph $G = (V, E)$, where $v_a \in V$ and $v_b \in V$ are two adjacent nodes (superpixels) having edge $e_{a,b} \in E, a \neq b$ between them, also the $n^{th}$ edge of $G$. The task is to label the binary indicator variable $y_{n;f} = I(x_{n;f} < \gamma_f)$ on edge $e_{a,b}$ such that $y_{n;f} = 1$ if $x_{n;f} < \gamma_f$, which means $v_{a;f}$ and $v_{b;f}$ are similar (belongs to the same proto-object), otherwise $y_{n;f} = 0$ if $v_{a;f}$ and $v_{b;f}$ are dissimilar (belongs to different proto-objects). Once $\gamma_f$ is computed, removing the edges such that $\mathbf{y}_f = 0$ results in isolated clusters of locally similar image patches, which are the desired groups of proto-objects.

Intuitively, any pair of adjacent nodes is either within the same proto-object cluster, or between different clusters ($y_{n;f} = \{1, 0\}$), therefore we consider two populations (the within-cluster edges, and the between-cluster edges) to be modeled from the density of $\mathbf{x}_f$ in a given image. In theory, this would mean that the density of $\mathbf{x}_f$ is a distribution exhibiting bi-modality, such that the left mode corresponds to the set of $\mathbf{x}_f$ that are considered similar and coherent, while the right mode contains the set of $\mathbf{x}_f$ that represent dissimilarity. At first thought, applying k-means with $k = 2$ or a mixture of two Gaussians would allow estimation of the two populations. However, there is no evidence showing that similarity distances follow symmetrical or normal distributions. In the following, we argue that the similarity distances $\mathbf{x}_f$ computed by EMD follow Weibull distribution, which is a distribution of the Exponential family that is skewed in shape.

We define $\text{EMD}(P, Q) = (\sum_i^m \sum_j^n f'_{ij} d_{ij})/(\sum_i^m \sum_j^n f'_{ij})$, with an optimal flow $f'_{ij}$ such that $\sum_j f'_{ij} \leq p_i$, $\sum_i f'_{ij} \leq q_j$, $\sum_{i,j} f'_{i,j} = \min(\sum_i p_i, \sum_j q_j)$, and $f'_{ij} \geq 0$, where $P = \{(x_1, p_1), ..., (x_m, p_m)\}$ and $Q = \{(y_1, q_1), ..., (y_n, q_n)\}$ are the two signatures to be compared, and $d_{ij}$ denotes a dissimilarity metric (i.e. $L_2$ distance) between $x_i$ and $y_j$ in $R^d$. When $P$ and $Q$ are normalized to have the same total mass, EMD becomes identical to Mallows distance [17], defined as $M_p(X, Y) = (\frac{1}{n} \sum_{i=1}^n |x_i - y_i|^p)^{1/p}$, where X and Y are sorted vectors of the same size, and Mallows distance is an $L_p$-norm based distance measurement. Furthermore, $L_p$-norm based distance metrics are Weibull distributed if the two feature vectors to be compared are correlated and non-identically distributed [7]. We show that our features assumptions are satisfied in Section 4.1. Hence, we can model each feature of $\mathbf{x}_f$ as a mixture of two Weibull distributions separately, and compute the corresponding $\gamma_f$ as the boundary locations between the two components of the mixtures. Although the Weibull distribution has been used in modeling actual image features such

as texture and edges [12][35], it has not been used to model EMD similarity distance statistics until now.

## 2.3 Weibull mixture model

Our Weibull mixture model (WMM) takes the following general form:

$$\mathcal{W}^K(\mathbf{x};\theta) = \sum_{k=1}^{K} \pi_k \phi_k(\mathbf{x};\theta_k) \ , \quad \phi(x;\alpha,\beta,c) = \frac{\beta}{\alpha}(\frac{x-c}{\alpha})^{\beta-1} e^{-(\frac{x-c}{\alpha})^{\beta}} \tag{1}$$

where $\theta_k = (\alpha_k, \beta_k, c_k)$ is the parameter vector for the $k^{th}$ mixture component, and $\phi$ denotes the three-parameter Weibull pdf with the scale ($\alpha$), shape ($\beta$), and location ($c$) parameter, and the mixing parameter $\pi$ such that $\sum_k \pi_k = 1$. In this case, our two-component WMM contains a 7-parameter vector $\theta = (\alpha_1, \beta_1, c_1, \alpha_2, \beta_2, c_2, \pi)$ that yields the following complete form:

$$\mathcal{W}^2(\mathbf{x};\theta) = \pi(\frac{\beta_1}{\alpha_1}(\frac{\mathbf{x}-c_1}{\alpha_1})^{\beta_1-1})e^{-(\frac{\mathbf{x}-c_1}{\alpha_1})^{\beta_1}} + (1-\pi)(\frac{\beta_2}{\alpha_2}(\frac{\mathbf{x}-c_2}{\alpha_2})^{\beta_2-1})e^{-(\frac{\mathbf{x}-c_2}{\alpha_2})^{\beta_2}} \tag{2}$$

To estimate the parameters of $\mathcal{W}^2(\mathbf{x};\theta)$, we tested two optimization methods: maximum likelihood estmation (MLE), and nonlinear least squares minimization (NLS). Both MLE and NLS requires an initial parameter vector $\theta'$ to begin the optimization, and the choice of $\theta'$ is crucial to the convergence of the optimal parameter vector $\hat{\theta}$. In our case, the initial guess is quite well defined: for any node of a specific feature $v_{j;f}$, and its set of adjacent neighbors $\mathbf{v}_{j;f}^N = N(v_{j;f})$, the neighbor that is most similar to $v_{j;f}$ is most likely to belong to the same cluster as $v_{j;f}$, and it is especially true under an over-segmentation scenario. Therefore, the initial guess for the first mixture component $\phi_{1;f}$ is the MLE of $\phi_{1;f}(\theta'_{1;f}; \mathbf{x}'_f)$, such that $\mathbf{x}'_f = \{\min(\text{EMD}(v_{j;f}, \mathbf{v}_{j;f}^N))|v_{j;f}; j = 1, ..., z, f \in \{i, c, o\}\}$, where $z$ is the total number of superpixels, and $\mathbf{x}'_f \subset \mathbf{x}_f$. After obtaining $\theta'_1 = (\alpha'_1, \beta'_1, c'_1)$, several $\theta'_2$ can be computed for the re-start purpose via MLE from the data taken by $Pr(\mathbf{x}_f|\theta'_1) > \mathbf{p}$, where $Pr$ is the cumulative distribution function, and $\mathbf{p}$ is a range of percentiles. Together, they form the complete initial guess parameter vector $\theta' = (\alpha'_1, \beta'_1, c'_1, \alpha'_2, \beta'_2, c'_2, \pi')$ where $\pi' = 0.5$.

### 2.3.1 Parameter estimation

Maximum likelihood estimation (MLE) estimates the parameters by maximizing the log-likelihood function of the observed samples. The log-likelihood function of $\mathcal{W}^2(\mathbf{x};\theta)$ is given by:

$$\ln \mathcal{L}(\theta;\mathbf{x}) = \sum_{n=1}^{N} \ln\{\pi(\frac{\beta_1}{\alpha_1}(\frac{x_n-c_1}{\alpha_1})^{\beta_1-1})e^{-(\frac{x_n-c_1}{\alpha_1})^{\beta_1}} + (1-\pi)(\frac{\beta_2}{\alpha_2}(\frac{x_n-c_2}{\alpha_2})^{\beta_2-1})e^{-(\frac{x_n-c_2}{\alpha_2})^{\beta_2}}\} \tag{3}$$

Due to the complexity of this log-likelihood function and the presence of the location parameters $c_{1,2}$, we adopt the Nelder-Mead method as a derivative-free optimization of MLE that performs parameter estimation with direct-search [22][16], by minimizing the negative log-likelihood function of Eq. 3.

For the NLS optimization method, first $\mathbf{x}_f$ are approximated with histograms much like a box filter that smoothes a curve. The appropriate histogram bin-width for data representation is computed by $w = 2(IQR)n^{-1/3}$, where IQR is the interquartile range of the data with $n$ observations [15]. This allows us to optimize a two component WMM to the height of each bin with NLS as a curve fitting problem, which is a robust alternative to MLE when the noise level can be reduced by some approximation scheme. Then, we find the least squares minimizer by using the trust-region method [27][28]. Both the Nelder-Mead MLE algorithm and the NLS method are detailed in the supplementary material.

Figure 2 shows the WMM fit using the Nelder-Mead MLE method. In addition to the good fit of the mixture model to the data, it also shows that the right skewed data (EMD statistics) is remarkably Weibull, this further validates that EMD statistics follow Weibull distribution both in theory and experiments.

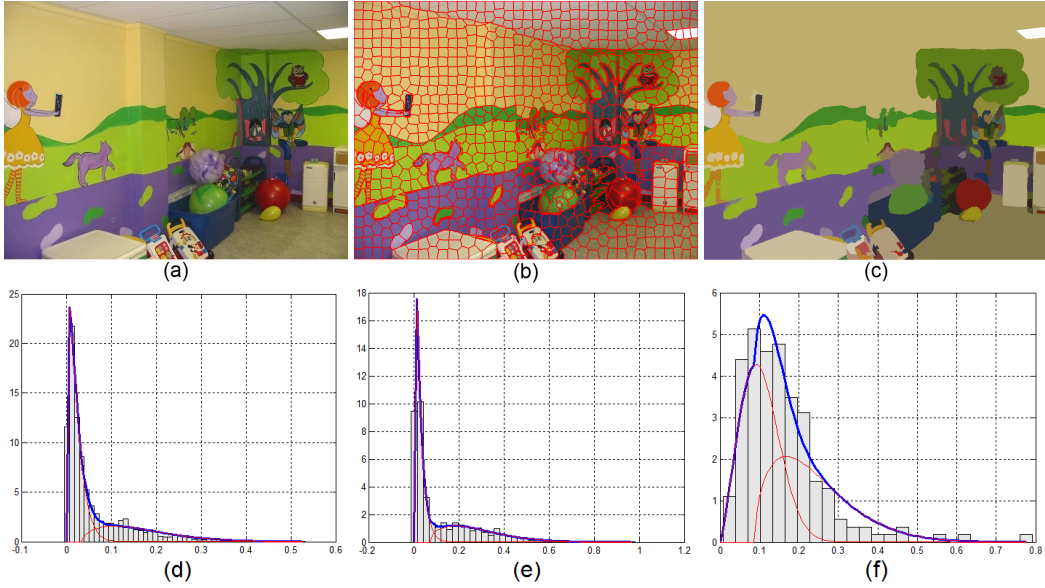

(a)                          (b)                          (c)

(d)                          (e)                          (f)

Figure 2: (a) original image, (b) after superpixel pre-processing [1] (977 initial segments), (c) final proto-object partitioning result (150 segments). Each final segment is shown with its mean RGB value to approximate proto-object perception. (d) $\mathcal{W}^2(\mathbf{x}_f; \theta_f)$ optimized using the Nelder-Mead algorithm for intensity, (e) color, and (f) orientation based on the image in (b). The red line indicates the individual Weibull components; and the blue line is the density of the mixture $\mathcal{W}^2(\mathbf{x}_f; \theta_f)$.

## 2.4 Visual clutter model with model selection

At times, the dissimilar population can be highly mixed in with the similar population, the density of which would resemble more of a single Weibull in shape such as Figure 2d. Therefore, we fit a single Weibull as well as a two component WMM over $\mathbf{x}_f$, and apply the Akaike Information Criterion (AIC) to prevent any possible over-fittings by the two component WMM. AIC tends to place a heavier penalty on the simpler model, which is suitable in our case to ensure that the preference is placed on the two-population mixture models. For models optimized using MLE, the standard AIC is used; for the NLS cases, the *corrected* AIC (AICc) for smaller sample size (generally when $n/k \leq 40$) with residual sum of squares (RSS) is used, and it is defined as $AICc = n\ln(RSS/n) + 2k + 2k(k+1)/(n-k-1)$, where $k$ is the number of model parameters, $n$ is the number of samples.

The optimal $\gamma_f$ can then be determined as follows:

$$\gamma_f = \begin{cases} \max(x, \epsilon), & \text{s.t. } \pi_1\phi_{1;f}(x|\theta_{1;f}) = \pi_2\phi_{2;f}(x|\theta_{2;f}) & \text{AIC}(\mathcal{W}^2) \leq \text{AIC}(\mathcal{W}^1) \\ \max(-\alpha_1(\ln(1-\tau))^{1/\beta_1}, \epsilon) & & \text{Otherwise} \end{cases} \quad (4)$$

The first case is when the mixture model is preferred, then the optimal $\gamma_f$ is the crossing point between the mixture components, and the equality can be solved in linear time by searching over the values of the vector $\mathbf{x}_f$; in the second case when the single Weibull is preferred by model selection, $\gamma_f$ is calculated by the inverse CDF of $\mathcal{W}^1$, which computes the location of a given percentile parameter $\tau$. Note that $\gamma_f$ is lower bounded by a tolerance parameter $\epsilon$ in both cases to prevent unusual behaviors when an image is nearly blank ($\gamma_f \in [\epsilon, 1]$), making $\tau$ and $\epsilon$ the only model parameters in our framework.

We perform Principle Component Analysis (PCA) on the similarity distance values $\mathbf{x}_f$ of intensity, color, and orientation and obtain the combined distance feature value by projecting $\mathbf{x}_f$ to the first principle component, such that the relative importance of each distance feature is captured by its variance through PCA. This projected distance feature is used to construct a minimum spanning tree over the superpixels to form the structure of graph $G$, which weakens the inter-cluster connectivity by removing cycles and other excessive graph connections. Finally, each edge of $G$ is labeled

according to Section 2.2 given the computed $\gamma_f$, such that an edge is labeled as 1 (similar) only if the pair of superpixels are similar in all three features. Edges labeled as 0 (dissimilar) are removed from $G$ to form isolated clusters (proto-objects), and our visual clutter model produces a normalized clutter measure that is between 0 and 1 by dividing the number of proto-objects by the initial number of superpixels such that it is invariant to different scales of superpixel over-segmentation.

# 3 Dataset and ground truth

Various in-house image datasets have been used in previous work to evaluate their models of visual clutter. The feature congestion model was evaluated on 25 images of US city/road maps and weather maps [25]; the models in [5] and [29] were evaluated on another 25 images consisting of 6, 12, or 24 synthetically generated objects arranged into a grid ; and the model from [18] used 58 images of six map or chart categories (airport terminal maps, flowcharts, road maps, subway maps, topographic charts, and weather maps). In each of these datasets, each image must be rank ordered for visual clutter with respect to every other image in the set by the same human subject, which is a tiring and time consuming process. This rank ordering is essential for a clutter perception experiment as it establishes a stable clutter metric that is meaningful across participants; alas it limits the dataset size to the number of images each individual observer can handle. Absolute clutter scales are undesirable as different raters might use different ranges on this scale.

We created a comparatively large clutter perception dataset consisting of 90 800×600 real world images sampled from the SUN Dataset images [33] for which there exists human segmentations of objects and object counts. These object segmentations serve as one of the ground truths in our study. The high resolution of these images is also important for the accurate perception and assessment of clutter. The 90 images were selected to constitute six groups based on their ground truth object counts, with 15 images in each group. Specifically, group 1 had images with object counts in the 1-10 range, group 2 had counts in the 11-20 range, up to group 6 with counts in the 51-60 range.

These 90 images were rated in the laboratory by 15 college-aged participants whose task was to order the images in terms of least to most perceived visual clutter. This was done by displaying each image one at a time and asking participants to insert it into an expanding set of previously rated images. Participants were encouraged to take as much time as they needed, and were allowed to freely scroll through the existing set of clutter rated images when deciding where to insert the new image. A different random sequence of images was used for each participant (in order to control for biases and order effects), and the entire task lasted approximately one hour. The average correlation (Spearman's rank-order correlation) over all pairs of participants was 0.6919 ($p < 0.001$), indicating good agreement among raters. We used the median ranked position of each image as the ground truth for clutter perception in our experiments.

# 4 Experiment and results

## 4.1 Image feature assumptions

In their demonstration that similarity distances adhere to a Weibull dstribution, Burghouts et al. [7] derived and related $L_p$-norm based distances from the statistics of sums [3][4] such that for non-identical and correlated random variables $X_i$, the sum $\sum_{i=1}^{N} X_i$ is Weibull distributed if $X_i$ are upper-bounded with a finite $N$, where $X_i = |s_i - T_i|^p$ such that $N$ is the dimensionality of the feature vector, $i$ is the index, and $s, t \in T$ are different sample vectors of the same feature.

The three image features used in this model are finite and upper bounded, and we follow the procedure from [7] with $L_2$ distance to determine whether they are correlated. We consider distances from one reference superpixel feature vector $s$ to 100 other randomly selected superpixel feature vectors $T$ (of the same feature), and compute the differences at index $i$ such that we are obtaining the random variable $X_i = |s_i - T_i|^p$. Pearson's correlation is then used to determine the relationship between $X_i$ and $X_j$, $i \neq j$ at a confidence level of 0.05. This procedure is repeated 500 times per image for all three feature types over all 90 images. As predicted, we found an almost perfect correlation between feature value differences for each of the features tested: Intensity: $100\%$, Hue: $99.2\%$, Orientation: $98.97\%$). This confirms that the low level image features used in this study follow a Weibull distribution.

| WMM-mle | WMM-nls | MS[9] | GB[11] | PL[6] | ED[19] | FC[25] | # Obj | C3[18] |
|---------|---------|-------|--------|-------|--------|--------|-------|--------|
| **0.8038** | **0.7966** | 0.7262 | 0.6612 | 0.6439 | 0.6231 | 0.5337 | 0.5255 | 0.4810 |

Table 1: Correlations between human clutter perception and all the evaluated methods. WMM is the Weibull mixture model underlying our proto-object partitioning approach, with both optimization methods.

## 4.2 Model evaluation

We ran our model with different parameter settings of $\epsilon \in \{0.01, 0.02, ..., 0.20\}$ and $\tau \in \{0.5, 0.6, ..., 0.9\}$ using SLIC superpixels [1] initialized at 1000 seeds. We then correlated the number of proto-objects formed after superpixel merging with the ground truth behavioral clutter perception estimates by computing the Spearman's Rank Correlation (Spearman's $\rho$) following the convention of [25][5][29][18].

A model using MLE as the optimization method achieved the highest correlation, $\rho = 0.8038$, $p < 0.001$ with $\epsilon = 0.14$ and $\tau = 0.8$. Because we did not have separate training/testing sets, we performed 10-fold cross-validation and obtained an average testing correlation of $r = 0.7599$, $p < 0.001$. When optimized using NLS, the model achieved a maximum correlation of $\rho = 0.7966$, $p < 0.001$ with $\epsilon = 0.14$ and $\tau = 0.4$, and the corresponding 10-fold cross-validation yielded an average testing correlation of $r = 0.7375$, $p < 0.001$. The high cross-validation averages indicate that our model is highly robust, and generalizable to unseen data.

It is worth pointing out that, the optimal value of the tolerance parameter $\epsilon$ showed a peak correlation at $0.14$. To the extent that this is meaningful and extends to people, it suggests that visual clutter perception may ignore feature dissimilarity on the order of $14\%$ when deciding whether two adjacent regions are similar and should be merged.

We compared our model to four other state-of-the-art models of clutter perception: the feature congestion model [25], the edge density method [19], the power-law model [6], and the C3 model [18]. Table 1 shows that our model significantly out-performed all of these previously reported methods. The relatively poor performance of the recent C3 model was surprising, and can probably be attributed to the previous evaluation of that model using charts and maps rather than arbitrary realistic scenes (personal communication with authors). Collectively, these results suggest that a model that merges superpixels into proto-objects best describes human clutter perception, and that the benefit of using a proto-object model for clutter prediction is not small; our model resulted in an improvement of at least $15\%$ over existing models of clutter perception. Although we did not record run-time statistics on the other models, our model, implemented in Matlab[1], had an end-to-end (excluding superpixel pre-processing) run-time of 15-20 seconds using $800 \times 600$ images running on an Win7 Intel Core i-7 computer with 8 Gb RAM.

## 4.3 Comparison to image segmentation methods

We also attempted to compare our method to state of the art image segmentation algorithms such as gPb-ucm [2], but found that the method was unable to process our image dataset using either an Intel Core i-7 machine with 8 Gb RAM or an Intel Xeon machine with 16 Gb RAM, at the high image resolutions required by our behavioral clutter estimates. A similar limitation was found for image segmentation methods that utilizes gPb contour detection as pre-processing, such as [8][14], while [23][34] took 10 hours on a single image and did not converge.

Therefore, we limit our evaluation to mean-shift [9] and Graph-based method [11], as they are able to produce variable numbers of segments based on the unsupervised partitioning of the 90 images from our dataset. Despite using the best dataset parameter settings for these unsupervised methods, our method remains the highest correlated model with the clutter perception ground truth as shown in Table 1, and that methods allowing quantification of proto-object set size (WMM, Mean-shift, and Graph-based) outperformed all of the previous clutter models .

We also correlated the number of objects segmented by humans (as provided in the SUN Dataset) with the clutter perception ground truth, denoted as # obj in Table 1. Interestingly, despite object

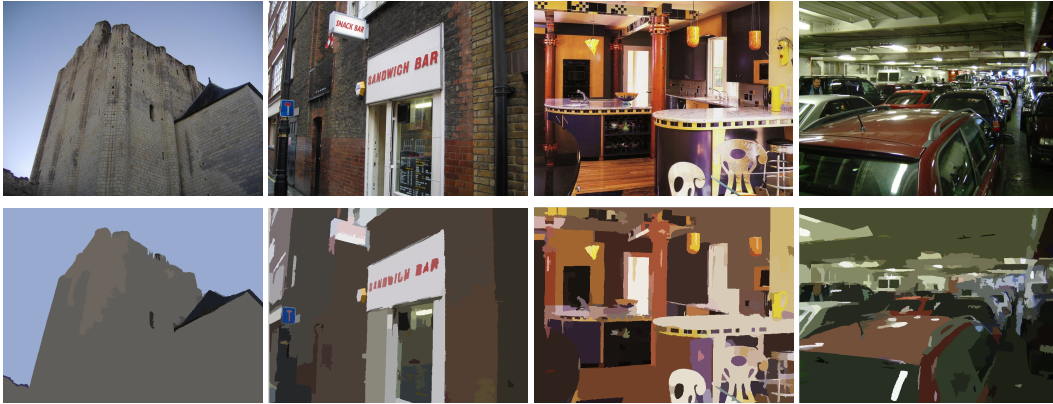

Figure 3: Top: Four images from our dataset, rank ordered for clutter perception by human raters, median clutter rank order from left to right: 6, 47, 70, 87. Bottom: Corresponding images after parametric proto-object partitioning, median clutter rank order from left to right: 7, 40, 81, 83.

count being a human-derived estimate, it produced among the lowest correlations with clutter perception. This suggests that clutter perception is not determined by simply the number of objects in a scene; it is the proto-object composition of these objects that is important.

## 5 Conclusion

We proposed a model of visual clutter perception based on a parametric image partitioning method that is fast and able to work on large images. This method of segmenting proto-objects from an image using mixture of Weibull distributions is also novel in that it models similarity distance statistics rather than feature statistics obtained directly from pixels. Our work also contributes to the behavioral understanding of clutter perception. We showed that our model is an excellent predictor of human clutter perception, outperforming all existing clutter models, and predicts clutter perception better than even a behavioral segmentation of objects. This suggests that clutter perception is best described at the proto-object level, a level intermediate to that of objects and features. Moreover, our work suggests a means of objectively quantifying a behaviorally meaningful set size for scenes, at least with respect to clutter perception. We also introduced a new and validated clutter perception dataset consisting of a variety of scene types and object categories. This dataset, the largest and most comprehensive to date, will likely be used widely in future model evaluation and method comparison studies. In future work we plan to extend our parametric partitioning method to general image segmentation and data clustering problems, and to use our model to predict human visual search behavior and other behaviors that might be affected by visual clutter.

## 6 Acknowledgment

We appreciate the authors of [18] for sharing and discussing their code, Dr. Burghouts for providing detailed explanations to the feature assumption part in [7], and Dr. Matthew Asher for providing their human search performance data on their work in Journal of Vision, 2013. This work was supported by NIH Grant R01-MH063748 to G.J.Z., NSF Grant IIS-1111047 to G.J.Z. and D.S., and the SUBSAMPLE Project of the DIGITEO Institute, France.

## Footnotes

[1]Code is available at mysbfiles.stonybrook.edu/~cheyu/projects/proto-objects.html

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
