[Supplementary Material]

# Appendix

**Chen-Ping Yu**
Department of Computer Science
Stony Brook University
cheyu@cs.stonybrook.edu

**Wen-Yu Hua**
Department of Statistics
Pennsylvania State University
wxh182@psu.edu

**Dimitris Samaras**
Department of Computer Science
Stony Brook University
samaras@cs.stonybrook.edu

**Gregory J. Zelinsky**
Department of Psychology
Stony Brook University
Gregory.Zelinsky@stonybrook.edu

## Abstract

Appendix for the details on Nelder-Mead algorithm, and the nonlinear least squares optimization method.

## 1 MLE using Nelder-Mead algorithm

Given a nondegenerate simplex $S$ as the convex hull of vertices $\{t_1, t_2, ..., t_{n+1}\} \in \mathbb{R}^n$, the Nelder-Mead method evaluates the objective function at each of the simplex vertices, the evaluation is denoted as $f(t_i)$, $i = 1, ..., n + 1$, $f(t) = -\ln \mathcal{L}(t; \mathbf{x})$. For each iteration, the algorithm consists of the following three steps [1]:

1. Order the evaluation values at each vertex, such that $f(t_1) \leq f(t_2) \leq \cdots \leq f(t_{n+1})$.
2. Calculate the centroid of the best n points by $c = \sum_{i=1}^{n} t_i$.
3. Compute the new simplex by finding a new accepted point $t'$ that leads to a better function evaluation. First, evaluate the *reflection* point $f(t_r)$ with respect to $c$. Iteration terminates if $f(t_1) \leq f(t_r) < f(t_n)$. Else:
   - *Expand* if reflected point is the new best point, such that $f(t_r) < f(t_1)$.
   - *Contract* if the reflected point is the second worse point, such that $f(x_n) \leq f(t_r)$.
   - *Shrink* the new simplex toward current best point $t_1$ if none of the above resulted in a better function evaluation, by replacing $t_i$ with $\frac{1}{2}(t_1 + t_i)$, for $i = 2, ..., n + 1$.

The effect of this algorithm is better understood in the case of $\mathbb{R}^2$, that the simplex is a triangle that flip-flops (reflection, if necessary) its way down the hill in the likelihood function space until convergence. To enforce the bound constraints of $\beta_{1,2} \geq 1$ and $\pi \leq 1$ that involves just inequalities, the accepted point $t'$ is adjusted with the respective lower and upper bound if any of its corresponding parameter values fail to satisfy the constraints.

## 2 Nonlinear least squares

For observations $(x_1, y_1), ..., (x_n, y_n)$ and our parameter vector $\theta = (\alpha_1, \beta_1, c_1, \alpha_2, \beta_2, c_2, \pi)$, the least squares estimator finds the minimizer to the following objective function:

$$F(\theta; \mathbf{x}) = \frac{1}{2} \sum_{i=1}^{n} r_i^2(\theta) \ , \qquad r_i(\theta) = \mathcal{W}^2(\theta; x_i) - y_i \tag{1}$$

where $r_i(\theta)$ is the residual, and $\theta$ is subject to the bound constraints that were specified earlier. The trust-region method iteratively minimizes the objective function starting from an initial starting parameter vector $\theta'$. By setting $\theta = \theta'$, we proceed to minimize a quadratic approximation $Q(s)$ that is the change in the objective function $F(\theta + s) - F(\theta)$. $Q(s)$ is given by:

$$Q(s) \equiv g^T s + \tfrac{1}{2} s^T H s \ , \quad \text{subject to} \quad ||s||_2 \leq \tau, \quad \tfrac{1}{\tau} - \tfrac{1}{||s||_2} = 0 \tag{2}$$

$s \in N$ is the subspace in the neighborhood $N$ of the trust-region, $g$ and $H$ are the gradient and Hessian at $\theta$. The Steihaug-Toint conjugate-gradient method can be used for each iteration step $s$ with the unconstrained Newton equation $Hs = -g$, setting $\theta = \theta + s$ if $F(\theta + s) < F(\theta)$ and adjusting $\tau$ at the end of each iteration [2][3]. In our constrained case, the unconstrained Newton step is replaced with a scaled Newton step $D^{-2}g = 0$ to solve for the following linear system:

$$MDs^N = -g', \quad M = D^{-1}HD^{-1} + diag(g)J^v \tag{3}$$

with $g' = D^{-1}g$ at the $k^{th}$ iteration, and $D$ is the diagonal matrix of vector $|v_k^{-1/2}|$, with $J^v$ denoting the Jacobian of $|v|$. The bound constraints ($ub$: upper-bound, $lb$: lower-bound) are used here for computing $v(\theta)$: for the $i^{th}$ observation, $v_i = \theta_i - ub_i$ if $g_i < 0$ and $ub_i < \infty$; $v_i = \theta_i - lb_i$ if $g_i \geq 0$ and $lb_i > -\infty$; $v_i = -1$ if $g_i < 0$ and $ub_i = \infty$; and $v_i = 1$ if $g_i \geq 0$ and $lb_i = -\infty$.