[Reviews · NeurIPS 2013]

Submitted by Assigned_Reviewer_2

UPDATED AFTER AUTHOR FEEDBACK AND OTHER REVIEWS:

Based on the author rebuttal and the comments of the other reviewers, I still believe the paper is worthy of acceptance. However, I agree with the other reviewers that the paper merits a ``1'' (incremental) rather than a ``2'' (major novelty) in the impact score.

ORIGINAL REVIEW:
Summary:
This paper proposes an image-based model for visual clutter perception (``a crowded, disorderly state''). For a given image, the model begins by applying an existing superpixel clustering then computing the intensity, colour and orientation histograms of pixels within each superpixel. Boundaries between adjacent superpixels are then retained or merged to create ``proto-objects''. The novel merging algorithm acts on the Earth Movers Distance (EMD), a measure of the similarity between two histograms. The distribution of histogram distances in each image for each image feature is modeled as a mixture of two Weibull distributions. The crossover point between the two distributions (or a fixed cumulative percentile if a single distribution is preferred by model selection) is used as the threshold point for merging: an edge is labelled ``similar'', and the superpixels merged, if the pair of superpixels exceed the threshold point for all three features. The clutter value for each image is the ratio of the final number of proto-objects to the initial number of superpixels (i.e. 0 = no proto-objects, not cluttered; 1 = all superpixels are proto-objects).
The model is validated by comparing to human clutter rankings of a subset of an existing image database. Human observers rank images from least to most cluttered, then the median ranking for each image is used as the ground truth for clutter perception. The new model correlates more highly with human rankings of clutter than a number of previous clutter perception and image segmentation models (including human object segmentation from a previous study).

Strengths:
In my opinion, scene clutter (a ``crowded, disorderly state'') is not the most interesting quality of perception, but a workable definition of ``set size'' in complex scenes is a very important problem. I think this paper is the precursor to an important contribution: namely, whether this clutter metric relates to performance in behavioural tasks such as search. The method for clustering seems to do a good job and to my knowledge is a sufficiently different approach to previous methods as to offer a novel contribution.

The behavioral ranking task is neat and the dataset will be of interest to the community for future model testing. The model correlates admirably with clutter rankings. I believe the paper to be a good contribution to the NIPS community.

Weaknesses:
It is not clear from the manuscript how strongly the success of the clutter metric depends on the type and parameters of the superpixel pre-processing. I assume this is done to make computation more tractable, but more detail should be provided in the manuscript as to why this does / does not matter.

The paper makes the theoretical (based on previous work) and empirical (based on three histograms) claim that the similarity distances follow Weibull distributions. I am not convinced beyond doubt that this is necessarily true. For example, I wonder whether the histograms in Figure 2 would be equally well fit by other mixture models using e.g. Gamma distributions. However, I don't think this distinction is important for the applicability of the method.

Clarity:
The manuscript is generally clearly written, but I believe some re-ordering of methodological details could save the reader some effort. For example, we are told in Section 2.1 that superpixels are extracted, but not until line 336 is the actual method referenced.
Line 149: ``However, there is no evidence showing that similarity distances follow symmetrical or normal distributions. Therefore, we argue that the similarity distances xf computed by EMD follow a Weibull distribution, which is a distribution of the Exponential family that is skewed in shape.'' The ``therefore'' implies that the Weibull distribution is a logical given, but it's not at all clear from this how a Weibull distribution was selected. The choice is justified more in subsequent paragraphs but the paper would be more clear if these paragraphs were rearranged: first tell us why you selected Weibull then tell us ``for these reasons we argue... Weibull''.

Minor points / suggestions:
-- for the initial MLE guess (line 190): I understand that this would be set separately for every image. Is this correct?

-- Line 269: ``making p and epsilon the only model parameters in our framework'': Should this read ``tau'', not ``p''? If not, what does ``p'' refer to here?

-- is it correct that each image will contain a potentially different threshold point, depending on the fitting of the mixture model to the distance histograms from that image?

-- section 4.1 is unclear to me: does ``image features'' here refer to the similarity distance metrics in each of intensity, color and orientation, or the intensity, color and orientation histograms themselves? In the latter case, is it right to say that a circular statistic such as orientation is ``upper bounded''?

-- lines 347--350: ``visual clutter perception may ignore feature dissimilarity on the order of 14% when deciding whether two adjacent regions are similar and should be merged.'' Given that the largest value of the epsilon parameter tested was 15%, I would want to see some dramatic drop of correlation within that 14--15% range to believe this more. A plot of the average correlations against epsilon (with five curves, one for each tau) would be informative here.

-- it is interesting that human object count is such a poor predictor of clutter. Perhaps the authors could provide example images from the database to show where clutter rankings and object counts diverge.

-- is it meaningful to ask why the correlations of the clutter model with the median ranking is higher than the inter-observer agreement? On line 309 you present the average correlation between pairs of participants. What is the correlation of each participant with the average of all other participants? That is, does the clutter model predict the average ranking better than any individual human?

-- the reader is often provided with an average rank order correlation and a p-value (presumably telling us that this correlation is significantly different to zero). I would find it more informative to be given the mean correlation and also the range of observed correlations.

-- First sentence of section 2.4 has an incorrect figure reference; should be Fig 2d?
-- Line 399 missing capital letter to start sentence.
-- Line 401 "our work suggests a mean" should be "means"

Summary: The paper presents a model of clutter perception using image feature distance metrics that is validated against human clutter rankings. The model and data set is novel enough to be of interest to the NIPS community.

Submitted by Assigned_Reviewer_5

This paper provides a new clutter quantifying method, which is still based on low level cues but uses intermediate level descriptors (proto-objects) between fragments and objects to capture the clutter in a scene.

The strength of the paper is that it is a complete work, from the definition of the problem to the presentation of the method, results and comparison to other methods. It is also a solid technical work that provides an algorithm that is superior to the other state-of-the-art clutter perception and image segmentation methods. In addition the paper is clearly written.

The main question I have is about originality of the work. The authors emphasize that the main original idea of the paper is the proposal that feature histogram similarity based on Earth Movers Distance measure should be modeled by the Weibull function. However, it seems to me that the Yanulevskaya & Geusebroek 2009 paper (which the authors do cite) essentially presents this idea. It is definitely the authors achievement that they put the idea into a working method, but they cannot claim full ownership of the original idea.

They also basically choose the next most obvious step to solve a problem and combine off-the-shelf methods to move to the next level. This is decent craftsmanship but it lends an incremental feel to the paper.

Typo: p. 5 last line: the first parameter is τ and not p.

Quality: Good.
Clarity: Good
Originality: Fair
Significance: Fair.
Summary: Good quality incremental craftsmanship.

Submitted by Assigned_Reviewer_8

This paper presents a method to estimate how much clutter there is in a given image. The method presented in the paper first extract superpixels from the image and then estimates the normalized Earth Mover's Distance (EMD) distance between neighboring superpixels with three modalities histograms - intensity, color and orientation. For each adjacent pair of superpixels it is estimated whether they are "similar" in which case they belong at the same cluster (called proto-object here) or "dissimilar" in which case they do not. The EMD distances distribution is modeled by a mixture of Weibull densities which is used to find the threshold of similarity. Finally, the measure for the clutterness is set to be the ratio of clusters (proto-objects) and the number of superpixels. The method is shown to perform well on a new dataset of clutter perception ground truth data.

Quality:
The suggested method is nice, though I feel the paper is not very interesting, not from the problem setting and not very a technical perspective. Much of the paper is devoted to how to learn a mixture of Weibull distributions, but there is not explanation as to why two methods are used, what are the fundamental differences (not the technical ones) and most importantly - how do they affect the final outcome. Moreover, I am wondering why EM was not used to estimate the mixture components (as it is the most popular choice in the literature). Furthermore, I have several concerns about the model evaluation section (4.2) and specifically - it is not mentioned if the model parameters were trained over a separate training set, or over the same images used as a test set (in which case the results are quite invalid).

Clarity:
THe paper is nicely written and presented

Originality and significant:
The contributions of this paper are somewhat underwhelming. Using Weibull statistics for EMD similarity distances is nice, but it is quite a small step forward, and I feel that the analysis of how to fit a mixture of Weibull densities is somewhat out of place for this paper (as I said before, it is technical and there is no analysis of how this affects the results other than numbers).


Summary: In summary, this is a nicely written paper which suffers from low significant and somewhat lacking analysis.

Submitted by Assigned_Reviewer_9

paper 116

This paper parses scenes into similar regions (called 'protoobjects') based on the similarity of their feature (color, intensity, orientation) histograms, and finds that the number of protoobjects discovered correlates with human rank orderings of visual clutter. This measure is appealing in that it is fairly intuitive and fast to compute, and the results (compared to other models) is impressive.

I think this paper misses the task dependence of visual clutter -- depending on what the visual search target is, different types of clutter will be more or less influential. A forest is a relatively uncluttered environment in which to search for a honda prius, but is very cluttered when searching for a pine tree (and vice versa for a parking lot). As I understand, visual clutter is a noteworthy measure insofar as it relates to search efficiency. I believe that the current measure of clutter might predict subjective rank orderings of clutter, but would fail to predict search efficiency because it does not respect the target-dependence of clutter measures.

Another intuitive measure of clutter is 'how easily could I insert an object that would be salient?' This is the intuition that ref [25] offers for their feature congestion model. Again, for this measure, counting the number of fairly homogenous regions in a display misses the degree to which some set of features would be salient in that image.

So, the current model seems very good at predicting subjective ratings of clutter, but I think it would be much worse than other clutter measures at predicting the magnitude of more practical implications of 'clutter' in a scene (which is what at least some of the prior models were designed to do).

At the very least, some discussion of this point is necessary. However, I think the current paper needs further experiments and measurements to assess whether the subjective ratings of clutter (which the current model predicts) have any relationship to the practical implications of clutter that researchers have previously been interested in.
Summary: The current proto-object parsing model predicts subjective measures of visual clutter, but I believe it is not well-suited to predict the practical implications of clutter: search difficulty and the ease of inserting a salient object into a scene. I think further experiments/measures are necessary to show that what the current model can predict relates meaningfully to practical consequences of visual clutter for search tasks.
Author Feedback

Author rebuttal: We thank the reviewers for the feedback. We will address as many of the comments as possible in the final copy.

First, we’d like to point out the technical significance of our proposed parametric modeling of Earth Mover's Distance (EMD) using Weibull distribution. Parametric models often suffer from difficulties in satisfying strict data assumptions, while non-parametric models do not. However, parametric models are more precise and efficient if the assumptions can be satisfied: we established the link between EMD and Weibull, and showed that our features satisfy the theoretical assumptions. Therefore, we believe it is also a more general contribution in all areas that require working with similarity distances, as we introduced a parametric way of working with the popular EMD similarity distance.

R2: We appreciate the positive comments and feedback. According to our experiments, different superpixel algorithms did not affect results much (except run time), while parameters are not an issue since we always used default settings.
Gamma may also be useful empirically (also a member of exponential family). We think it is more theoretically appropriate to use Weibulls due to the connection that we have established between Weibull and Lp-distance statistics, on the specifics of [7]. Furthermore, we showed that our features do satisfy the self-correlated and non-identically distributed assumptions which justified the connection (Sec. 4.1).
Minor points:
1. The automatic initial MLE guesses are different for every image, but based on the same method (line 184~195).
2. Should be “tau”, thanks!
3. Yes, each image contains a different similarity threshold, automatically computed.
4. Image features in Section 4.1 refer to the actual intensity, color, and orientation histograms. Upper-bounded means that each feature’s values have a finite upperbound (intensity: 255, hue: 1, orientation: 359).
5. The correlation steadily drops further from r=0.8094 to r=0.7572 as epsilon goes from 0.14 to 0.2, at a 0.01 interval. We will try to make space for this plot.
6. Our intuition was that human object count does not account for complex vs simple objects, and thus is not a reliable set-size measure.
7. True, but it is not unprecedented as [25] also reported their model (r=0.83) correlated higher than inter-subject correlation (r=0.72).
8. Aside from the full range of parameters, it would be more meaningful to compute mean correlation from more appropriate subset of parameter values, i.e. tau=0.6~0.9, epsilon=0.05~0.15 results in mean r=0.7298; or r=0.7787 when tau=0.6~0.9 and epsilon=0.10~0.15.

R5: It is crucial to make the distinction between actual image statistics, and similarity distance statistics (line 165~167), which are two very different quantities. Here we stress our use of Weibull to model EMD (Lp-based similarity distance) being completely different from using Weibull to model actual image features (whole image and local patch edge distributions) as done in [35]: similarity distance values are the results of computing the similarity between pairs of actual feature distributions. We provided the theoretical justification for using Weibull to model EMD (Section 2.2 and 4.1), hence Weibull is the appropriate distribution for EMD. This conclusion is derived from a theoretical basis that is completely different from image features being Weibull ([35]). This theoretical justification of parametric modeling similarity distance is novel to the best of our knowledge.

R8: We point the Reviewer to the comments of R2 on why the problem setting is interesting. Regarding the technical contributions:
1. MLE and NLS are two different, widely-used classes of optimization methods; our motivation was to test how different optimization methods affect the final model predictions. Both methods result in high correlation of our model to human clutter rating (table 1), indicating that our approach is very robust and mostly invariant to different optimization methods.
2. EM is a general class of methods for finding the MLE, where the M step usually requires taking the derivative of the likelihood function. In our case, the likelihood function of the 7-parameter Weibull mixture is very complex (Eq. 3) and taking the gradient of such complex likelihood function is often avoided (Kaylan and Harris, 1982, Mumford 1997). Therefore, we adopted the derivative-free method of Nelder-Mead algorithm for obtaining the MLE. In fact, the algorithm is very similar to EM in that its step 1 and 2 (Sup Material Sec. 1) is essentially the E-step, while step 3 is the M-step in a non-derivative way. Fig 2 shows that the Nelder-Mead algorithm worked well empirically.
3. We have validated in TWO ways: We followed the convention for reporting the optimal dataset parameters and correlation by other state of the art clutter models ([6][19][25][18]) using the entire dataset (r=0.8094). We also applied 10-fold cross validation (line 340~345) resulted in an average unseen test-set correlation of r=0.7891, such high correlation indicates that our model is very generalizable and robust to unseen data.

R9: The goal of our work is to propose a quantification of set-size that can be used in modeling clutter perception. It is the first step leading up to a larger exploration of the relationship between clutter and search behavior as the first Reviewer (R2) also points out. On the other hand, Asher et al, Jrnl of Vision 2013 recently found that existing clutter models ([6][19][25]) were rather weak in predicting search performance using 120 natural scenes (r~0.2), with mostly non-significant p-values. Our preliminary experiments using Asher’s human search dataset resulted in r=0.29 and p=0.0015, which is the highest correlation with the most significant p comparing to existing clutter models that Asher reported. We anticipate conducting a similar study to evaluate our model in this regard in a follow-up journal paper.